# Effects of Reduced Phosphate Fertilizer and Increased Trichoderma Application on the Growth, Yield, and Quality of Pepper

**DOI:** 10.3390/plants12162998

**Published:** 2023-08-19

**Authors:** Xiaoyu Duan, Chunlei Zou, Yifan Jiang, Xuejing Yu, Xueling Ye

**Affiliations:** 1Key Laboratory of Protected Horticulture of Education Ministry and Liaoning Province, College of Horticulture, Shenyang Agricultural University, National and Local Joint Engineering Research Centre of Northern Horticultural, Facilities Design and Application Technology (Liaoning), Shenyang 110866, Chinajiangyifan1024@163.com (Y.J.); yxj529@stu.syau.edu.cn (X.Y.); 2Vegetable Research Institute of Liaoning Academy of Agricultural Sciences, Shenyang 110161, China; zouchunlei2008@163.com

**Keywords:** phosphorus, *Trichoderma*, pepper, plant growth, fruit quality, nutrient uptake

## Abstract

Phosphorus utilization by crop plants is often limited, thereby resulting in large accumulations of residual phosphorus fertilizer in the soil. *Trichoderma* fungi function as natural decomposition agents that can contribute to increasing decomposition and promoting nutrient absorption in plants. In this study, we developed a novel fertilizer application strategy that reduces phosphate fertilizer and increases *Trichoderma* and examined its effects on the growth, nutrient absorption, and fruit quality of pepper (*Capsicum annuum* L.). We compared the efficacies of eight treatments: P100 = standard dose application of phosphorus fertilizer; P85 = 85% dose; P70 = 70% dose; P0 = no phosphorus fertilizer; and the TP100, TP85, TP70, and TP0 treatments, in which a *Trichoderma* mixture was added to the P100, P85, P70, and P0 treatments, respectively. The combined fertilizer application strategy stimulated plant growth, increased chlorophyll content, improved yield, and enhanced nutrient absorption. Additionally, the strategy improved pepper fruit quality by increasing the contents of soluble proteins, soluble sugars, vitamin C, capsaicin, and capsanthin. A comprehensive analysis indicated that the TP85 treatment was the optimal fertilization regime for pepper. This study provides a novel fertilizer application strategy for pepper that not only ensures good plant growth but also protects soil health.

## 1. Introduction

Given its unique flavor and high nutritional value, pepper (*Capsicum annuum* L.) is a commercially important crop that is extensively cultivated worldwide [1]. The flavors of the pepper fruit are diverse, ranging from sweet (e.g., sweet pepper) to hot and pungent (e.g., chili pepper), thus providing consumers with an abundance of culinary options [2]. In terms of nutritional value, the pepper fruit is rich in micronutrients and bioactive compounds, including vitamin C, carotenoids, and phenolic compounds, which may contribute to preventing various diseases, including diabetes, cancer, and coronary disorders [3]. At the mature stage, the colors of pepper fruit flesh can range from green to yellow and red, with differences in pigmentation being attributed to differences in the accumulation of chlorophylls, carotenoids, and capsanthin, respectively [4,5]. The quality of pepper fruit quality is prominently determined by factors such as fertilizer application, irrigation, climate, and soil properties. For example, the provision of Ca(NO_3_)_2_ has been found to enhance the vitamin C and carotenoids contents of sweet pepper [6]. Furthermore, the application of humic acid has been shown to promote increases in the antioxidant constituents of pepper fruit, including total flavonoids, capsaicin, lycopene, and β-carotene [7]. Under drought conditions, the application of K^+^-rich carrot compost can modify the accumulation of carotenoids and phenolic compounds in pepper [8]. In addition, an appropriate ratio of ammonium-nitrate (25:75) not only promotes root development but also improves the content of micronutrients and bioactive compounds in pepper fruits [9], whereas nutritional solutions with high K^+^ concentrations (400 and 500 mg·L^−1^) are beneficial for pepper yield and quality [10].

Phosphorus is considered to be one of the most important macronutrients for improving plant growth and development, including root development, stem thickening, and flower formation [11]. This element is an essential component of nucleic acids, nucleotides, and enzymes in cells that play roles in numerous physiological and biochemical processes in plants, including photosynthesis and nitrogen (N) fixation [12,13]. Given its insoluble nature and the fact that it is often fixed in soil, the availability of phosphorus in the soil is generally limited. Consequently, to sustain the productivity and quality of plants, moderate amounts of inorganic phosphorus fertilizer are typically applied to provide sufficient amounts of this nutrient for plants [14]. However, high amounts of phosphorus fertilizer can induce the rapid combination between phosphorus and aluminum (Al), iron (Fe), and calcium (Ca), thereby forming complexes in which this element is unavailable for uptake by plant roots [15]. The fixation of phosphorus in soil is influenced by a range of factors, including soil pH, temperature, moisture, and the presence of other soil minerals. Moreover, the continuous application of phosphorus fertilizer results in a reduction in soil fertility, changes in microbial diversity, and lower plant yields [15]. Consequently, identifying environmentally friendly and effective alternative strategies is necessary to enhance the availability of phosphorus fertilizers for plants.

Fungi in the genus *Trichoderma* have been widely used in agriculture as biocontrol agents and biofertilizers. Species of *Trichoderma* colonize plant roots and therein develop mutually beneficial relationships with their host plants [16]. In doing so, they not only contribute to the development of disease resistance but also improve the quality and yield of crops. As a biocontrol agent, the novel *Trichoderma asperellum* has been shown to significantly inhibit the mycelial growth of the pathogenic *Fusarium oxysporum lycopersici*, and its high levels of chitinase and β-1,3-glucanase activities also suppress the growth of this pathogen. Accordingly, it can be used as a commercial biocontrol agent to control tomato wilt [17]. When infected by *Colletotrichum truncatum,* chili peppers treated with *Trichoderma* have been found to be characterized by a higher accumulation of phenols and higher expressions of defense-related genes, thereby indicating that *Trichoderma* could enable plants to prime their defense networks against phytopathogens [18]. Additionally, *Trichoderma* could activate the plant-mediated resistance mechanism in response to *Plasmopara viticola* sporangia germination to improve immunity against *P. viticola* and reduce downy mildew severity in grapevines [19]. As a biofertilizer, *Trichoderma* can be applied to plants in different ways, including the inoculation of soil and as a foliar spray, which can promote increases in yield and nutrient uptake, and enhance fruit quality [20,21]. For example, in tomatoes, the application of *Trichoderma* was found to enhance total yield and significantly improve fruit quality, including the contents of lycopene, asparagine, GABA, and MEA [22]. Similarly, Chinese cabbage treated with *Trichoderma* as a biofertilizer for 30 days has been shown to have significantly improved productivity and quality, as a consequence of enhanced nutrient uptake and the alleviation of environmental stress [23]. Furthermore, in grapes, the secondary metabolites of *Trichoderma* can induce disease resistance, promote plant growth, and increase polyphenol contents and antioxidant activity [24]. Additionally, *Trichoderma* has been demonstrated to enhance the potential solubilization of phosphate to further improve soil fertility and promote mangrove growth [25]. Similarly, *Trichoderma*-enriched bio-organic fertilizer has been found to increase the abundance of soil microflora and enhance soil fertility, thereby improving tomato fruit quality [26].

The continuous application of synthetic fertilizers can result in the degradation of soil and perturbation of the rhizosphere microbiota, and also affects soil fertility. Given the aforementioned properties of *Trichoderma*, it is reasonable to assume that these problems could be solved through a combination of the enhanced application of *Trichoderma* and a reduction in the application of synthetic fertilizers, which would not only contribute to the sustainable development of soil but also reduce the harmful effects of excessive fertilizer application on the production and quality of plants. In this study, we sought to identify a sustainable and environmentally friendly method for fertilizer application to pepper plants. To this end, we applied different combinations of phosphorus fertilizer and *Trichoderma* and investigated the influence of the combined application of these two agents on the growth and development, fruit quality, and nutrient components in peppers grown under field conditions.

## 2. Results

### 2.1. Effects of Phosphorus Fertilizer and Trichoderma Applications on Plant Growth

In this study, pepper samples were collected after different treatments at two different stages (the green and red fruit stages) (Figure 1). Different plant growth indices were measured, including plant height, stem diameter, chlorophyll content (SPAD), and the dry weights of different tissues. The results showed that during the green fruit stage, plant heights, SPAD values, and the dry weights of roots and leaves differed significantly among plants receiving treatments with the different levels of phosphorus fertilizer. Furthermore, we detected significant differences in plant heights, stem diameters, SPAD values, and the dry weights of roots, stems, leaves, and fruits following exposure to applied *Trichoderma*, whereas only the dry weights of fruits showed significant differences following treatment with a combination of the different phosphorus fertilizer levels and *Trichoderma* application. During the red fruit stage, the dry weights of different tissues showed significant differences in response to different phosphorus fertilizer levels and *Trichoderma* alone, respectively, whereas the dry weights of stems and leaves showed significant differences in response to combined phosphorus fertilizer and *Trichoderma* treatments (Appendix A). Further analysis revealed that the combination of a *Trichoderma* mixture with phosphorus fertilizer could stimulate plant growth. For example, at the green fruit stage, plant heights in the TP100 treatment and stem diameters in the TP0 treatment were higher than those in the P100 and P0 treatments, respectively. However, plant height and stem diameters at the red fruit stage were not significantly affected by *Trichoderma* application (Figure 2a,b). At the green fruit stage, chlorophyll contents (SPAD) in the TP100, TP85, and TP70 treatments were significantly higher than those in the P100, P85, and P70 treatments, respectively. In addition, plants in the TP85 treatment group exhibited higher chlorophyll content during the red fruit stage (Figure 2c).

We also found that the dry weights of different tissues were enhanced by the addition of *Trichoderma* to different levels of phosphorus fertilizers. At the green fruit stage, the dry weights of roots in the TP85 treatment were significantly higher than those in the P85 treatment, whereas at the red fruit stage, the dry weights of roots in the TP0 treatment were significantly higher than those in the P0 treatment (Figure 3a). Compared with those in the P100, P70, and P0 treatments, we detected differences in the dry weights of stems in the TP100, TP70, and TP0 treatments, respectively, although only at the red fruit stage (Figure 3b). In addition, the dry weights of leaves in the TP85 treatment were significantly higher than those in the P85 treatment, and the dry weights of fruits in the TP70 and TP0 treatments were significantly higher than those in the P70 and P0 treatments at the green fruit stage (Figure 3c,d). Moreover, we found that the co-application of *Trichoderma* and phosphorus fertilizer contributed to increases in the dry weights of leaves and fruits at the red fruit stage (Figure 3c,d). Collectively, these findings thus indicate that the addition of *Trichoderma* to phosphorus fertilizer treatments at different concentrations contributed to an enhancement of plant growth.

### 2.2. Effects of Phosphorus Fertilizer and Trichoderma Applications on Plant Nutrition

To assess the effects of phosphorus fertilizer and *Trichoderma* application on plant nutrient accumulation, we examined the nutrient contents of whole plants. The results indicated that there were significant differences in the N, P, and K contents of whole plants with respect to the different treatment parameters, namely, the different levels of phosphorus fertilizer, the application of *Trichoderma*, and the interaction of different phosphorus fertilizer levels and *Trichoderma* application at two different stages of plant growth (Appendix A). During the green fruit stage, the N content of whole plants subjected to the TP85 and TP0 treatments were significantly higher than those in the P85 and P0 treatments, respectively (Figure 4a). Similar trends were identified with respect to the levels of P and K in plants, in that the contents of these elements in whole plants in the TP85, TP70, and TP0 treatment groups were significantly higher than those in the P85, P70, and P0 groups, respectively. Furthermore, the highest whole-plant contents of P and K were detected in those plants subjected to the TP85 treatment (Figure 4b,c). During the red fruit stage, the N contents of whole plants in the TP100, TP85, and TP70 treatment groups were significantly higher than those in the P100, P85, and P70 groups, respectively (Figure 4a). Furthermore, in all treatments including *Trichoderma,* we found that the whole-plant contents of P and K were significantly higher than those in plants subjected to the corresponding *Trichoderma*-free treatments. (Figure 4b,c). Moreover, we established that plants subjected to the TP85 treatment were characterized by the highest uptake of N, P, and K. On the basis of these results, it can thus be inferred that the application of *Trichoderma* promoted plant nutrient absorption, with the best effects being observed in those plants in the TP85 treatment group.

### 2.3. Effects of Phosphorus Fertilizer and Trichoderma Applications on Pepper Fruit Quality

To gain a further insight into the factors associated with the influence of the combined phosphorus fertilizer–*Trichoderma* treatments on pepper fruit quality, we examined a range of relative indices, including the contents of soluble proteins, soluble sugars, vitamin C, capsaicin, and capsanthin (Figure 5). The results provided evidence to indicate that treatment with different levels of phosphorus fertilizer led to significant differences in the contents of vitamin C, soluble sugars, capsaicin, and capsanthin during the green fruit stage, whereas the application of *Trichoderma* contributed to significant differences in the contents of soluble proteins, vitamin C, and capsanthin; and the combined application phosphorus fertilizer at different levels and *Trichoderma* led to significant differences in the contents of vitamin C, capsaicin, and capsanthin. During the red fruit stage, we detected significant differences in all assessed indices in response to the different phosphorus fertilizer levels and *Trichoderma* addition, whereas the contents of vitamin C, capsaicin, and capsanthin showed significant differences in response to the interactions between the different levels of phosphorus fertilizer and *Trichoderma* application (Appendix A).

At the green fruit stage, the soluble protein content of TP0-treated plants was found to be considerably higher than that of the P0-treated plants, whereas at the red fruit stage, the soluble protein content of plants in the TP100 and TP85 treatment groups was notably higher than that in plants receiving the P100 and P85 treatments, respectively (Figure 5a). Whereas we found that the application of *Trichoderma* in conjunction with phosphorus fertilizer had no significant effect on the soluble sugar content of pepper fruits at the green fruit stage, at the red fruit stage, the soluble sugar content of fruits was significantly elevated in response to the TP100 and TP0 treatments (Figure 5b). With respect to the vitamin C contents of pepper fruit, we found that supplementation of phosphorus fertilizer with *Trichoderma* led to an increase in this vitamin at the green fruit stage in plants subjected to the TP100, TP85, and TP70 treatments. Similarly, the vitamin C contents of pepper fruits at the red fruit stage were significantly higher in plants receiving the TP85 and TP70 treatments than in those plants in P85 and P70 treatment groups, respectively (Figure 5c). Furthermore, compared with the P85 treatment, the capsanthin content of pepper fruits at the green fruit stage was found to be significantly higher after the TP85 treatment, whereas the capsanthin content of fruits at the red fruit stage was higher after the TP85 and TP70 treatments compared with that after the P85 and P70 treatments, respectively (Figure 5d). Similarly, the capsaicin content of pepper fruits at the green fruit stage was found to be significantly higher following the supplementation of phosphorus fertilizer with *Trichoderma* compared with the non-supplemented application of phosphorus fertilizer. Likewise, at the red fruit stage, the capsaicin content of fruits was significantly higher after the TP100 and TP85 treatments than that after the P100 and P85 treatments, respectively (Figure 5e). Overall, we established that the TP85 treatment showed the best results in terms of pepper fruit quality.

### 2.4. Effects of Phosphorus Fertilizer and Trichoderma Applications on Yield per Plant

We found that treatment with different levels phosphorus fertilizer, *Trichoderma* addition, and their interactions all had an influence on the yield per plant measured at the mature stage (Table 1), with the highest values being recorded among those plants in the TP85 treatment group (Figure 6). Furthermore, in all cases, the yields of plant receiving treatments involving the application of *Trichoderma* were considerably higher than those receiving the corresponding treatments without *Trichoderma* supplementation, which implies that the application of *Trichoderma* can contribute to improvements in pepper yield (Figure 6).

### 2.5. Effects of Phosphorus Fertilizer and Trichoderma Applications on Soil Fertility

Analysis of the fertility of soil in the different treatment plots following treatment revealed that during the green fruit stage, the application of different levels of phosphorus fertilizer led to significant differences in the total P, available P, total K, and available K contents of soil. Furthermore, inoculating soil with *Trichoderma* led to significant differences in the total K and available K contents and the combined phosphorus fertilizer–*Trichoderma* treatments led to significant differences in the total N, alkali-hydrolyzed N, and available K contents. During the red fruit stage, with the exception of total K content, we detected significant differences in the content of other nutrients in plots receiving inputs of the different levels of phosphorus fertilizer. Moreover, we detected significant differences in the contents of total and available P in response to the addition of *Trichoderma*. With the exception of the contents of alkali-hydrolyzed N, those of other nutrients showed significant differences attributable to the interaction between the different levels of phosphorus fertilizer and *Trichoderma* application (Appendix A).

Detailed analysis of soil fertility revealed that compared with the TP0 treatment, the total N content of soils at the green fruit stage was significantly higher in the P0 treatment. At the red fruit stage, the addition of *Trichoderma* resulted in an increase in the total N contents after the TP100 and TP85 treatments compared with those after the P100 and P85 treatments, respectively (Figure 7a). Similar results were obtained for the contents of alkali-hydrolyzed N at the green fruit stage, with contents measured in response to the P0 treatment being higher than those recorded after the TP0 treatment (Figure 7b). We also found that the addition of *Trichoderma* to the different phosphorus fertilizer treatments resulted in differences in the total P content. For example, during the green fruit stage, the total P content was observed to show opposite trends in response to the TP100 and TP70 treatments, with the total P content being lower after the TP100 treatment but higher after the TP70 treatment compared with that after the corresponding control treatments. Moreover, at the red fruit stage, we detected reductions in total P content in response to the TP100 and TP85 treatments, whereas an increase in contents was recorded following the TP0 treatment, compared with the corresponding control treatment (Figure 7c). At the green fruit stage, we detected considerably lower levels of available P in response to the TP70 treatment compared with those measurement after the P70 treatment. Similar results were obtained at the red fruit stage for the TP70 and P70 and the TP0 and P0 treatments, whereas contrasting patterns were observed for the TP100 and P100 treatments (Figure 7d). At the green fruit stage, the application of *Trichoderma* was found to elevate the total K content for the TP0 treatment and similar elevations were observed following the TP100 and TP85 treatments at the red fruit stage, compared with the corresponding control treatment (Figure 7e). Likewise, at the green fruit stage, increases in available K contents were detected in response to the P100, P85, and P0 treatments, although at the red fruit stage increases were only in plots subjected to the P100 treatment (Figure 7f). On the basis of these finding, we can infer that the supplementation of phosphate fertilizer with *Trichoderma* can contribute to promoting availability of soil nutrient for plant absorption.

## 3. Discussion

Phosphorus has been established to be an essential nutrient for maintaining plant growth and development and plays important roles in multiple physiological processes, including those associated with nutrition and photosynthesis [27]. Although the phosphorus fertilizer content of soil is generally high, phosphorus is readily fixed within the soil, thereby limiting its absorption by plants [28]. Furthermore, it has been demonstrated that excessive application of phosphorus fertilizers can actually reduce plant yield and exacerbates the risk of soil pollution [29]. Fungi in the genus *Trichoderma* play important roles in the decomposition of organic matter, thereby enhancing the availability of nutrients in the soil [30,31]. Consequently, in this study, we sought to validate our assumption that a combined fertilization strategy based on a reduction in phosphorus fertilizer application and supplementation with an optimal concentration of *Trichoderma* may provide a more effective approach to minimizing the likelihood of fertilizer pollution in soil, whilst also ensuring adequate plant growth. To assess the efficacy of this fertilizer application strategy on plant growth, we examined a range of physiological indices, including plant height, stem diameter, and the dry weights of different tissues. The results revealed that although the supplementation of different amounts of phosphorus fertilizer with *Trichoderma* had no appreciable effects on the heights or stem diameters of pepper plants at the red fruit stage (Figure 2a,b), the chlorophyll content (SPAD) of pepper leaves was significantly influenced by the addition of *Trichoderma* at the two different growth stages we assessed (Figure 2c). Leaf chlorophyll content is generally considered a reliable indicator of the nutrient status of plants [32] and is closely associated with photosynthetic capacity. Accordingly, the higher chlorophyll content we detected in the leaves of pepper plants in response to the application of *Trichoderma* is considered to indicate that the treated plants had sufficient nutritional resources and a relatively strong photosynthetic capacity. Furthermore, compared with those of plants treated with the application of phosphorus fertilizer alone, we found that the dry weights of the roots, stems, leaves, and fruits of pepper plant were significantly higher after the application of phosphorus fertilizer in combination with *Trichoderma* (Figure 3a–d). These results accordingly indicate that the addition of *Trichoderma* can contribute to the healthier growth of peppers plants.

To examine the effects of *Trichoderma* supplementation of different phosphorus fertilizer treatments on plant nutrient absorption, we measured the whole-plant contents of N, P, and K, as well as the levels of these elements in soil. Compared with the application of phosphorus fertilizer alone, we found that plants receiving the combined application of phosphorus fertilizer and *Trichoderma* were characterized by considerably higher whole-plant contents of N, P, and K, and that among the assessed treatments, the TP85 treatment contributed to the highest contents of N, P, and K at the whole-plant level (Figure 4a–c). In contrast, we detected reductions in the N, P, and K contents of soil following the addition of *Trichoderma* (Figure 7a–f). Consistent with our observations, it has previously been found that the application of *Trichoderma* can contribute to root development by enhancing the growth of root and root tips [33]. Similarly, the application of *Trichoderma* has been demonstrated to increase the areas and lengths of cucumber roots, as well as promoting significant increases in the content of nutrients such as Cu, P, Fe, and Zn in *Trichoderma*-inoculated plants [34]. Consequently, our observations of higher N, P, and K contents in pepper plants and lower contents in soils following the application of phosphorus fertilizer combined with *Trichoderma* may be explained by the fact that the application of *Trichoderma* enhances plant growth by facilitating the absorption of nutrients from the soil.

To further determine the effects of *Trichoderma* application on pepper fruit quality, we examined a range of relevant plant indices. Briefly, with the exception of soluble sugars, we found that the application of *Trichoderma* contributed to inducing the accumulation of soluble proteins, vitamin C, capsaicin, and capsanthin at the green fruit stage. Similarly, pepper quality, which is determined to a large extent by the contents of soluble proteins, soluble sugars, vitamin C, capsaicin, and capsanthin, was significantly elevated at the red fruit stage in response to the application of *Trichoderma* (Figure 5a–e). Additionally, *Trichoderma* supplementation was demonstrated to promote a significant improvement in pepper fruit yield, with the TP85 treatment showing the best results in this regard (Figure 6). Collectively, our findings in this study revealed that the supplementation of phosphorus fertilizer with *Trichoderma* can significantly influence plant physiology. Similar effects have been observed for other horticulture crops. For example, application of *Trichoderma* as a biofertilizer has been shown to enhance nutrient uptake and improve the quality and productivity of flowering Chinese cabbage plants [23]. Similarly, the application of *Trichoderma gamsii* It-62 has been shown to enhance the capacity of common beans to solubilize tricalcium phosphate, thereby further improving plant growth, P uptake, and total protein content [35]. In addition, *Trichoderma* application has been found to enhance fruit quality and yield in tomatoes and grapes [24,36]. In this context, it is worth mentioning that *Trichoderma* also plays an important role in influencing the response of plants to abiotic stress. For example, volatile organic compounds associated with *Trichoderma* have been demonstrated to enhance the tolerance of *Arabidopsis thaliana* to salt stress [37]. Moreover, it has been found that *Trichoderma* RM-28 can reduce the pH and electrical conductivity of red mud (which is highly alkaline and saline), thereby contributing to an improvement in the growth of sorghum and Sudangrass seedlings grown in this soil, presumably by alleviating oxidative stress [38]. In addition, recent studies have reported that plants treated with *Trichoderma* show numerous changes in the levels of mRNAs and proteins involved in plant defense responses, signal transduction, and fruit quality improvement [39,40,41]. However, in the present study, we did not specifically examine differences in the levels of genes/proteins involved in secondary metabolite pathways or seek to characterize nutritional crosstalk networks. The specific mechanisms underlying the relationships among nutrient absorption, yield enhancement, and fruit quality improvement thus warrant further analysis.

## 4. Material and Methods

### 4.1. Plant Material and Experimental Conditions

This study was conducted in a research field at the Liaoning Academy of Agricultural Sciences in Shenyang (41° N, 123° E), Liaoning, China. The region has a temperate semi-humid continental climate with four distinct seasons. During the growing season in the present study, we recorded an average temperature and average precipitation of 19.8 °C and 598.7 mm, respectively. Prior to planting the experimental plants, we analyzed the physical and chemical properties and nutrient status of the soil in the research field, the results of which are presented in Table 2. A red pepper cultivar, obtained from the Liaoning Academy of Agricultural Sciences, was sown in plugs in early March 2019, and upon the emergence of three true leaves, the seedlings were transplanted into the prepared field. For each treatment, we allocated a plot area of 11.25 m, with each treatment consisting of three replicates. In each treatment plot, a total of 32 red pepper seedings were grown.

### 4.2. Experimental Design and Treatments

For the purposes of this study, we assessed the efficacy of eight experimental treatments. The P100, P85, P70, and P0 treatments involved the application of a standard dose of phosphorus fertilizer, 85% of the standard dose, 70% of the standard dose, and no application of phosphorus fertilizer, respectively (Table 3). The TP100, TP85, TP70, and TP0 treatments entailed the addition of a *Trichoderma* mixture to phosphorus fertilizer at the four corresponding dosage levels (P100, P85, P70, and P0, respectively).

The *Trichoderma* mixture used was in the form of a powder (CFU·g^−1^: 1 × 10^9^) provided by the College of Plant Protection, Heilongjiang Bayi Agricultural University (Daqing, China). This powder was mixed at a 1:22 (g/g) ratio with a sample soil that had been sieved through a size 10 mesh. A 10-g sample of the *Trichoderma* mixture was applied to each plant in the relevant treatments. To ensure healthy growth, a topdressing was applied during pepper growth and development. Diseases and insect pests were controlled simultaneously during routine management.

### 4.3. Measurement of Plant Growth Indices

Plant height and stem diameter were measured using a flexible rule (Deli, Ningbo, China) and electronic vernier calipers (Deli, Ningbo, China), respectively. The chlorophyll content of the leaves was measured using an automatic soil plant analysis development (SPAD) meter (CL-01; Hansatech, Norfolk, UK), with SPAD readings being obtained from the center of leaves (avoiding the mid-vein) [42]. To determine dry weights, plants were harvested from the research field and immediately washed. Subsequently, roots, stems, and leaves were separated, oven-dried at 105 °C for 30 min, and then further oven-dried at 65 °C until obtaining constant weights. All samples were weighed using an electronic balance (BSA224S; Sartorius, Gottingen, Germany).

### 4.4. Determination of the N, P, and K Content of Pepper Plants

For each treatment, we collected a number of pepper plants, from which the roots, stems, leaves, and fruits were separated, oven-dried at 105 °C for 30 min, and then further oven-dried at 70 °C until obtaining constant weights. The different tissues were crushed and weighed for each analysis (total N, P, and K content). The total plant contents of N, P, and K were determined using an automatic Kjeldahl distillation-titration unit [43], a molybdate colorimetric method [44], and a sodium hydroxide flame photometer [45], respectively. The nutritional elements of the entire plants are represented as the sum of these elements in the different tissues.

### 4.5. Determination of Fruit Yield and Quality Indexes

Red pepper yield was measured using an electronic balance. Soluble sugar content was measured following a previously described method [46]. Soluble protein content and vitamin C levels were measured using the Coomassie Brilliant blue method [47] and the 2,6-dichloroindophenol stain method [48], respectively. For the determination of capsaicin, 5 g fruit from each treatment was collected and oven-dried at 65 °C. Thereafter, the samples were placed in 50-mL centrifuge tubes, to which was added 25 mL of extraction solution (methyl alcohol:tetrahydrofuran = 1:1). The tubes were placed in a 60 °C water bath for 30 min and extracts (containing capsaicin) were obtained by ultrasound disruption. The extracts were filtered through filter paper and collected in clean centrifuge tube. The filter papers and residues were again placed in 50-mL centrifuge tubes, to which 25 mL of the extraction solution was added. This process was performed twice. The extracts obtained from the three extractions were combined and thoroughly mixed. The composite solutions were then dried using a concentration vacuum centrifuge, dissolved in 10 mL of extraction solution, and filtered through a 0.22-μm nylon membrane. The capsaicin content was analyzed using a high-performance liquid chromatograph (Waters 2695-series HPLC; Waters Corp., Milford, MA, USA) equipped with a UV-detector (Waters e2489) and C18 column (250 mm × 4.6 mm, 5 μm, Waters, Massachusetts, USA). The mobile phase consisted of methyl alcohol:water (65:35, *v*/*v*). Operating conditions were as follows: a column temperature of 30 °C, flow rate of 1.0 mL·min^−1^, and injection volume of 10 μL. Capsaicin content was measured at 280 nm and quantified using a capsaicin standard and standard curves. For the extraction and measurement of capsanthin, 0.3 g of pepper fruit was ground into a powder and placed in a clean flask. Having added 80 mL of acetone, capsanthin was extracted for 20 min. The mixture was filtered into a fresh volumetric flask and the final volume was adjusted to 100 mL. Subsequently, 1 mL of the extract was collected in a 10-mL volumetric flask, and the volume was adjusted to 10 mL for capsanthin measurement. Capsanthin was measured using an ultraviolet spectrophotometer, with the detection wavelength set at 460 nm, optical length at 10 nm, slit width at 1 nm, response time at 2 s, and scanning speed at 480 nm·min^−1^.

### 4.6. Determination of the N, P, and K Content of Soil

At five locations within each treatment plot. soil samples were collected to a depth of 20 cm using a soil sampler, and thoroughly mixed to give a composite sample for each plot. Subsequently, the soil samples were air-dried in the laboratory, and sieved through 0.85-mm and 0.15-mm meshes for the determination of N, P, and K contents. The total N and alkali-hydrolyzed N contents in soil were determined using an automatic Kjeldahl distillation-titration unit and an alkali-hydrolyzation diffusion method, respectively [49,50]. The total P content in soil was determined using a molybdate colorimetric method, and the available P content was determined using sodium bicarbonate extraction and molybdenum blue colorimetry [51]. The total K and available K contents in soil were determined using a sodium hydroxide flame photometer [52]. All analyses were performed using three biological replicates.

### 4.7. Data Analysis

All assays were conducted with at least three biological replicates. The data were analyzed using one-way and two-way analyses of variance (ANOVA) to determine significant differences based on treatment, with Duncan’s and LSD multiple range tests being used to evaluate significant differences between treatments (*p* < 0.05). All analyses were performed using SPSS software (23.0 version, Chigaco, IL, USA).

## 5. Conclusions

In this study, we showed that the addition of a *Trichoderma* mixture, in conjunction with moderate reductions in the levels of applied phosphorus fertilizer, could contribute to promoting the growth, nutrient absorption, and yield of pepper plants grown in soils that receive average applications of phosphorus fertilizer throughout the year. Furthermore, we established that the combined application of phosphorus fertilizer and *Trichoderma* can improve pepper fruit quality by enhancing the contents of soluble sugars, soluble proteins, vitamin C, capsanthin, and capsaicin. Integrating these results, we identified the TP85 treatment (application of an 85% standard dose of phosphorus fertilizer with a *Trichoderma* mixture) as the optimum fertilizer application strategy. This combinational strategy is not only beneficial in terms of enhancing pepper plant quality and yield but may also contribute to the sustainable development of healthy soil and reductions in the input costs incurred by farmers.

## Figures and Tables

**Figure 1 plants-12-02998-f001:**
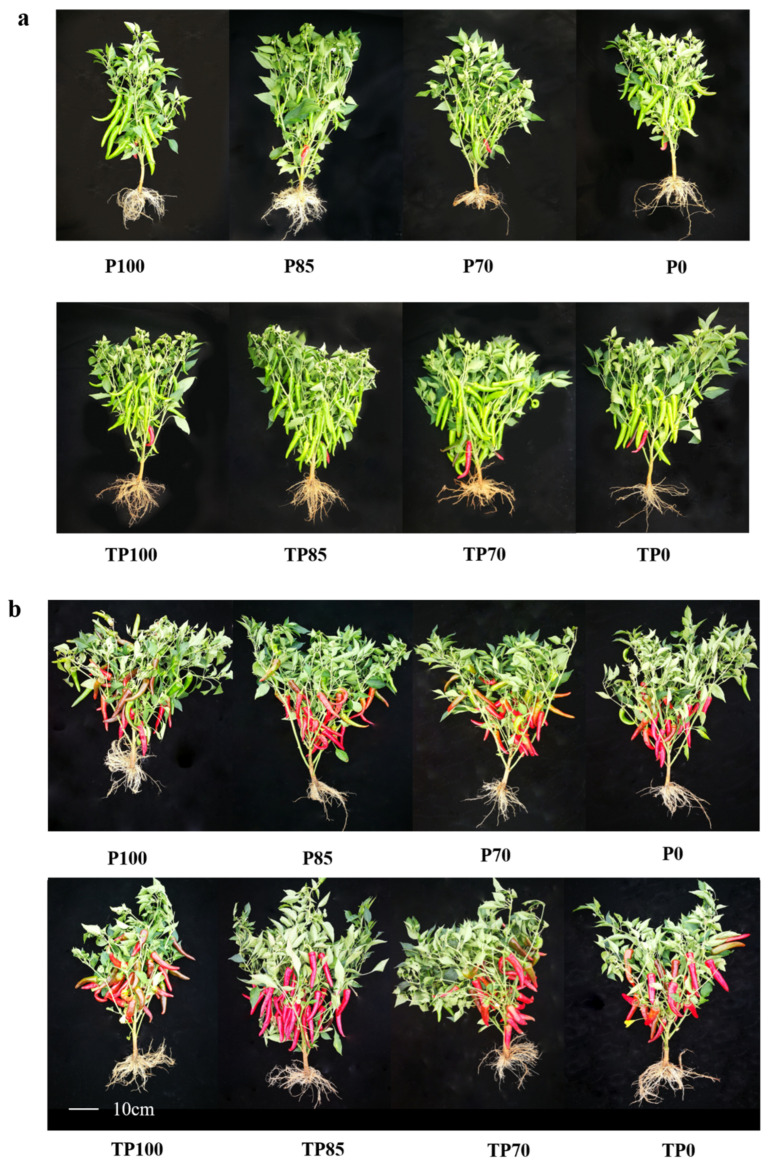
Plant characteristics of different levels of phosphate fertilizer without and with *Trichoderma* at two different fruit developmental stages. (**a**) Plant characteristics of different levels of phosphate fertilizer without and with *Trichoderma* at green fruit stage. (**b**) Plant characteristics of different levels of phosphate fertilizer without and with *Trichoderma* at red fruit stage.

**Figure 2 plants-12-02998-f002:**
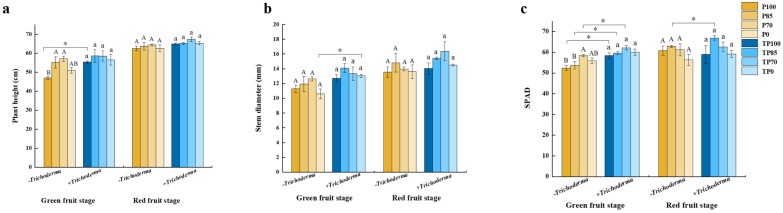
The effect of different levels of phosphate fertilizer and *Trichoderma* on plant growth. (**a**) The effect of different levels of phosphate fertilizer and *Trichoderma* on plant height. (**b**) The effect of different levels of phosphate fertilizer and *Trichoderma* on stem diameter. (**c**) The effect of different levels of phosphate fertilizer and *Trichoderma* on SPAD. The data were analyzed from three biological replicates. The different uppercase letters represent significant difference among different level of phosphate fertilizer treatment groups without *Trichoderma*; the different lowercase letters represent significant difference among different level of phosphate fertilizer treatment groups with *Trichoderma*; the asterisk represent significant difference between comparison with and without *Trichoderma* at the same level of phosphate fertilizer, * *p* < 0.05.

**Figure 3 plants-12-02998-f003:**
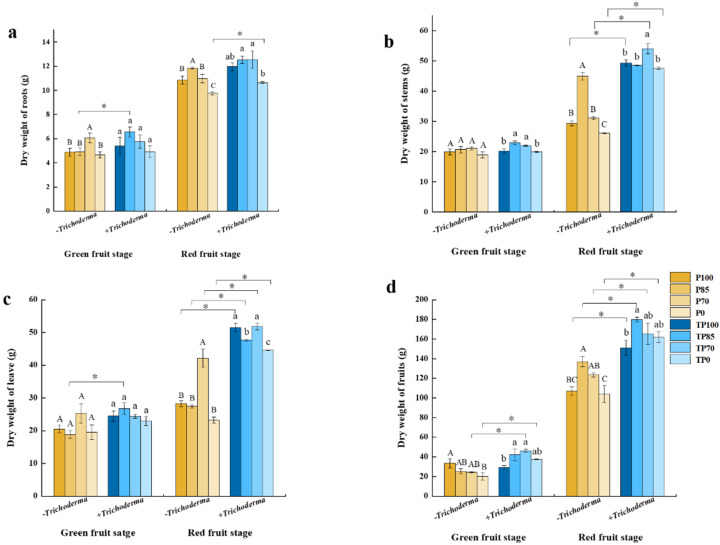
The effect of different levels of phosphate fertilizer and *Trichoderma* on the dry weight of different tissues. (**a**) The effect of different levels of phosphate fertilizer and *Trichoderma* on the dry weight of roots. (**b**) The effect of different levels of phosphate fertilizer and *Trichoderma* on the dry weight of stems. (**c**) The effect of different levels of phosphate fertilizer and *Trichoderma* on the dry weight of leave. (**d**) The effect of different levels of phosphate fertilizer and *Trichoderma* on the dry weight of fruits. Three biological replicates for each treatment, and the data were analyzed. The different uppercase letters represent significant difference among different level of phosphate fertilizer treatment groups without *Trichoderma*; the different lowercase letters represent significant difference among different levels of phosphate fertilizer treatment groups with *Trichoderma*; the asterisk represent significant difference between comparison with and without *Trichoderma* at the same level of phosphate fertilizer, * *p* < 0.05.

**Figure 4 plants-12-02998-f004:**
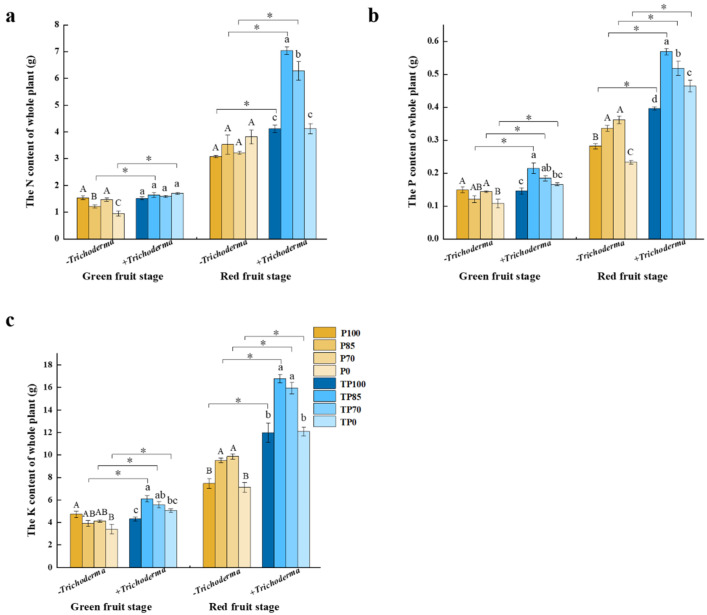
The effect of different levels of phosphate fertilizer and *Trichoderma* on plant nutrition accumulation. (**a**) The effect of different levels of phosphate fertilizer and *Trichoderma* on the total N content of whole plant. (**b**) The effect of different levels of phosphate fertilizer and *Trichoderma* on the total P content of whole plant. (**c**) The effect of different levels of phosphate fertilizer and *Trichoderma* on the total K content of whole plant. Three biological replicates for each treatment, and the data were analyzed. The different uppercase letters represent significant difference among different levels of phosphate fertilizer treatment groups without *Trichoderma*; the different lowercase letters represent significant difference among different levels of phosphate fertilizer treatment groups with *Trichoderma*; the asterisk represent significant difference between comparison with and without *Trichoderma* at the same level of phosphate fertilizer, * *p* < 0.05.

**Figure 5 plants-12-02998-f005:**
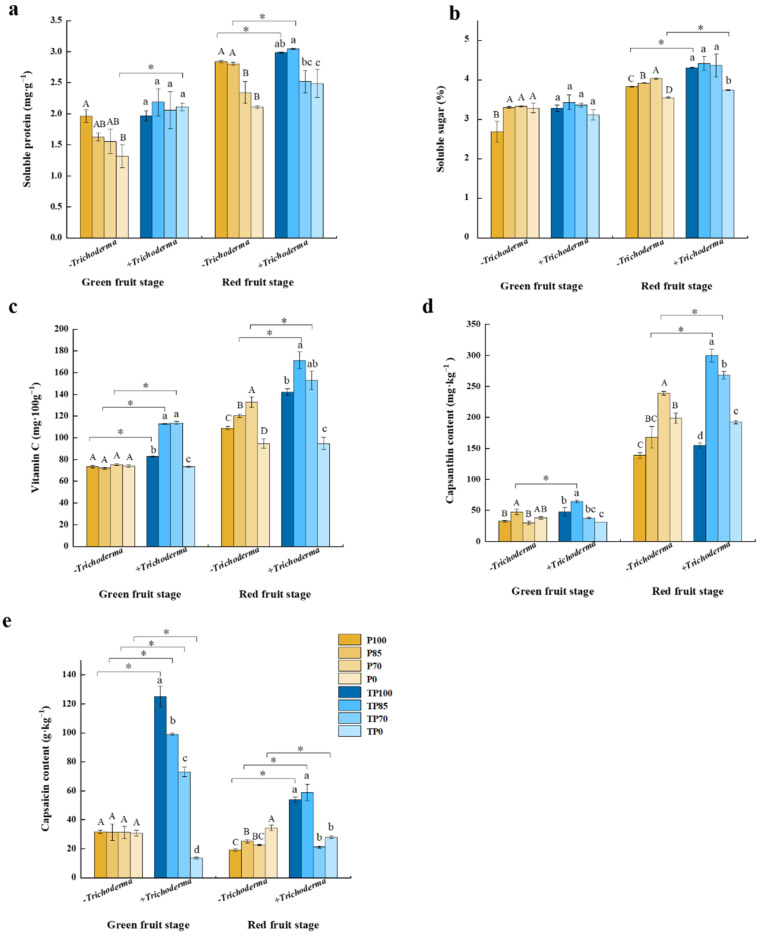
The effect of different levels of phosphate fertilizer and *Trichoderma* on the pepper fruit quality at different development stages of pepper fruit. (**a**) The effect of different levels of phosphate fertilizer and *Trichoderma* on the soluble protein content. (**b**) The effect of different levels of phosphate fertilizer and *Trichoderma* on the soluble sugar content. (**c**) The effect of different levels of phosphate fertilizer and *Trichoderma* on the VC content. (**d**) The effect of different levels of phosphate fertilizer and *Trichoderma* on the capsanthin content. (**e**) The effect of different levels of phosphate fertilizer and *Trichoderma* on the capsaicin content. Three biological replicates for each treatment, and the data were analyzed. The different uppercase letters represent significant difference among different levels of phosphate fertilizer treatment groups without *Trichoderma*; the different lowercase letters represent significant difference among different levels of phosphate fertilizer treatment groups with *Trichoderma*; the asterisk represent significant difference between comparison with and without *Trichoderma* at the same level of phosphate fertilizer, * *p* < 0.05.

**Figure 6 plants-12-02998-f006:**
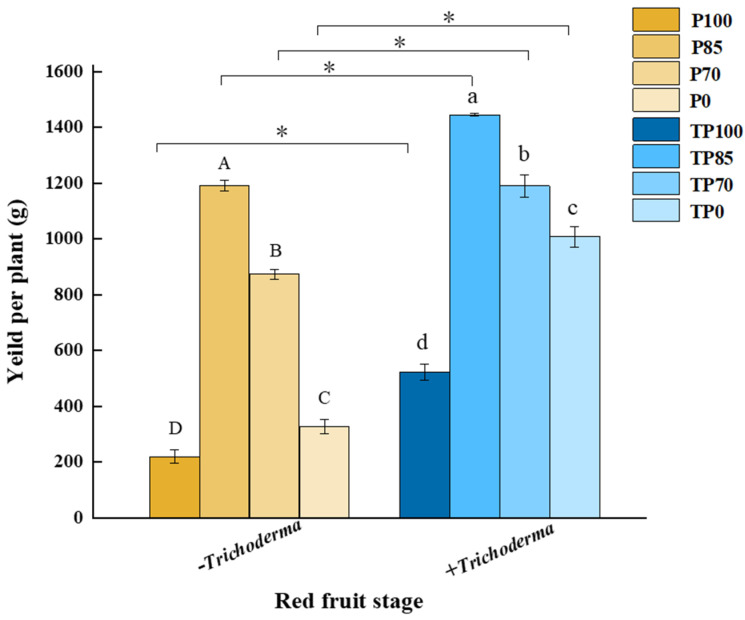
The effect of different levels of phosphate fertilizer and *Trichoderma* on the yield per plant in red fruit stage. Three biological replicates for each treatment, and the data were analyzed. The different uppercase letters represent significant difference among different levels of phosphate fertilizer treatment groups without *Trichoderma*; the different lowercase letters represent significant difference among different levels of phosphate fertilizer treatment groups with *Trichoderma*; the asterisk represent significant difference between comparison with and without *Trichoderma* at the same level of phosphate fertilizer, * *p* < 0.05.

**Figure 7 plants-12-02998-f007:**
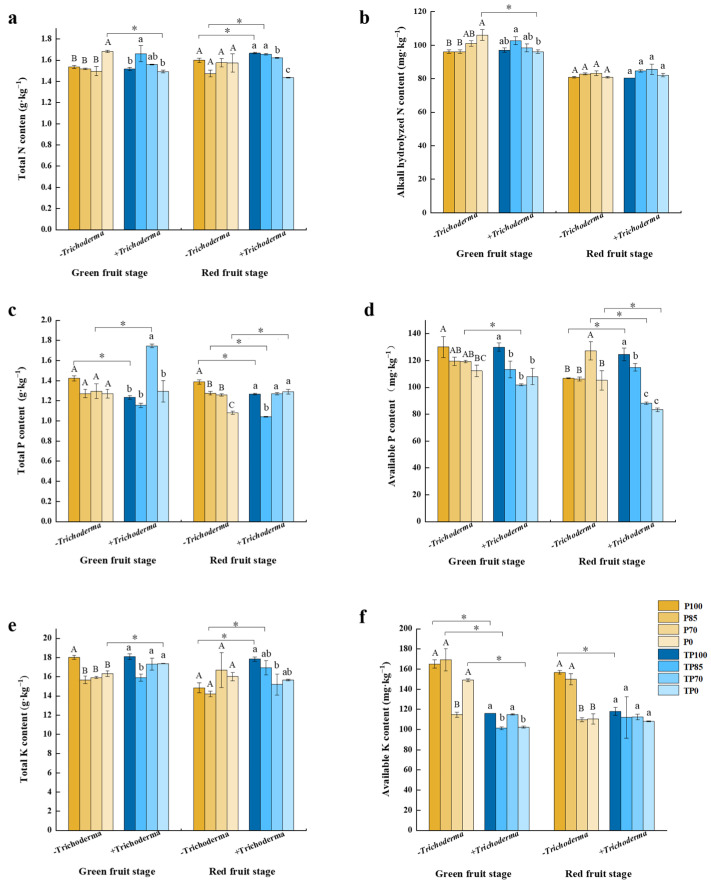
The effect of different levels of phosphate fertilizer and *Trichoderma* on the soil nutrients at different development stages of pepper fruit. (**a**) The effect of different levels of phosphate fertilizer and *Trichoderma* on the total N content in soil. (**b**) The effect of different levels of phosphate fertilizer and *Trichoderma* on the alkali hydrolyzed N content in soil. (**c**) The effect of different levels of phosphate fertilizer and *Trichoderma* on the total P content in soil. (**d**) The effect of different levels of phosphate fertilizer and *Trichoderma* on the available P content in soil. (**e**) The effect of different levels of phosphate fertilizer and *Trichoderma* on the total K content. (**f**) The effect of different levels of phosphate fertilizer and *Trichoderma* on the available K content in soil. Three biological replicates for each treatment, and the data were analyzed. The different uppercase letters represent significant difference among different levels of phosphate fertilizer treatment groups without *Trichoderma*; the different lowercase letters represent significant difference among different levels of phosphate fertilizer treatment groups with *Trichoderma*; the asterisk represent significant difference between comparison with and without Trichoderma at the same level of phosphate fertilizer, * *p* < 0.05.

**Table 1 plants-12-02998-t001:** The effect of different phosphate fertilizer levels, *Trichoderma*, and their interaction on yield per plant.

Red Fruit Stage	Different Levels of Phosphate		*Trichoderma*		Different Levels of Phosphate * *Trichoderma*	
	*F*	*P*	*F*	*P*	*F*	*P*
Yield per plant	492.556	***	432.774	***	27.692	***

Note: * *p* < 0.05; *** *p* < 0.001.

**Table 2 plants-12-02998-t002:** Properties of the experimental soil.

pH	EC (μm·g^−1^)	Total N Content (g·kg^−1^)	Total P Content (g·kg^−1^)	Total K Content (g·kg^−1^)	Alkali Hydrolyzed N Content (mg·kg^−1^)	Available P Content (mg·kg^−1^)	Available K Content (mg·kg^−1^)	Organic Matter (%)
6.64	217.82	1.81	1.18	10.54	102.28	103.97	81.6	4.45

**Table 3 plants-12-02998-t003:** The amount of fertilizer application for each community.

	Base Fertilizer (g)			Top Dressing (g)		
Treatment	Calcium Superphosphate	Urea	Potassium Sulfate	Calcium Superphosphate	Urea	Potassium Sulfate
P100	562.50	146.74	135.00	168.84	178.92	40.50
P85	478.08	146.74	135.00	143.51	178.92	40.50
P70	393.84	146.74	135.00	118.19	178.92	40.50
P0	0.00	146.74	135.00	0.00	178.92	40.50

## Data Availability

All relevant data are within the manuscript.

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
