# Peer review of "Effects of Reduced Phosphate Fertilizer and Increased Trichoderma Application on the Growth, Yield, and Quality of Pepper"

_plants, 2023, doi:10.3390/plants12162998_

Round 1

Reviewer 1 Report

The limitation of phosphorus utilization in production is widespread, resulting in a large amount of residual phosphorus fertilizer in the soil. Trichoderma is a natural decomposing agent that can promote nutrient absorption in plants. This study proposes a new compound fertilization strategy that reduces phosphorus fertilizer and increases Trichoderma, exploring its effects on plant growth, nutrient absorption, and fruit quality. It provides a new fertilization strategy for chili peppers that can ensure plant growth and protect soil health. The main modification suggestions are as follows:

Introduction:

1.     Line 37-41, “Pepper fruit quality is affected by distinctive factors, such as fertilizer, irrigation, climate, and soil properties. For example, foliar application of salicylic acid can significantly improve pepper vegetative growth, fruit quality, and yield [6]. Light as an important factor positively affect pigment accumulation in bell peppers [7].” These sentences had little relevance to the context. Here should emphasis and discuss the benefits of applying some nutrients to pepper, used to lead to the following text.

2.     Line 65, suggest adding a summary sentence to introduce the two main functions of Trichoderma in the following text: it not only can resist diseases effectively but also improve the quality and yield of crops.

3.     The introduction section does not provide sufficient background information. It was mentioned that the application of phosphorus fertilizer and Trichoderma can affect soil fertility in result 2.5, so the introduction section needs further overview.

Results:

1.     Figure 1: should add the ruler.

2.     Line 126: “the P85 and P70 treatments at the green fruit stage.” This sentence should be added the data in the figure 3 above. Such as (Figure 3c, d).

3.     Line 186: “The yield per plant was the highest in TP85 treatment.” This sentence should be added the data in the figure 6 above. Such as (Figure 6).

4.     Line 214: There is an additional period (.) after the word “stage”.

5.     Line 222: The results here are well described, but a lack of summary sentence like the previous results.

6.     Insufficient explanation of experimental results. Result “2.5 Effects of Phosphorus Fertilizer and Trichoderma Applications on Soil Fertility” In this result, there are inconsistencies in different stage in green fruit stage and red fruit stage, how to explain this result?

Such as in Figure 6c, in green fruit stage, the total P content of TP70 treatment was increased after adding Trichoderma, but in red fruit stage cannot found this trend; and in Figure 6d, The P100 and P85, TP100 and TP85 treatment, the changing trends during the two fruit development stages are also different; and in Figure 6e, how can explain there are nothing similarity in the changes in K content during the two fruit development stages between different treatments?

And whether this result is directly related to the protection of soil health mentioned in line 25 of the abstract still needs further verification. Therefore, it is necessary to further clarify the results of 2.5 points and consider the word of the abstract.

Discussion:

1.     Line 227: There is an additional period (.) after the word “fertilizer”.

2.     Line 239: Missing comma (,) before “and”.

Conclution:

1.     The summary of the article is not comprehensive enough. Conclution needs more in it, as it's more of an afterthought. The authors are suggested to highlight important findings and include afterthought of this work.

2.     The contribution of the article is not obvious enough. The significance of this paper is not expounded sufficiently. The author needs to highlight this paper's innovative contributions.

Author Response

Dear reviewer, 

We sincerely thanks for you taking time to carefully review our manuscript and put out valuable comments for improving this manuscript. After carefully studying these comments, we have revised the manuscript followed by your suggestions. We hope that this revised manuscript could make your satisfaction.

Best wishes!

Xiaoyu Duan 

Reviewer 2 Report

Abstract should also include the different treatments of the experiment; in addition, there is a mistake with the term 'nutrition absorption': the correct is 'nutrient absorption'. Please, correct it throughout the whole text. 

Paragraph 2.1 is very extended; please, decrese its length, by focusing only on the most important data of the experiment; avoid redundant information, making the text monotonous.

In the title of the paragraph 2.2 the term 'plant nutrition accumulation' is incorrect. Please, convert it either into: 'plant nutrition' or into: 'nutrient accumulation'. 

Description of data in the paragraph 2.3 is also quite monotonous; please revise and focus only on the most important results. 

In the title of the paragraph 2.5 the term 'soil nutrition' is also incorrect. Revise into: 'soil fertility'. This paragraph is also (as the previous ones) written in a monotonous style... 

In the Table 1 I do not see any physical properties... Please, revise the title into: 'Properties of the experimental soil'

The information provided in the paragraph 4.2 for the treatments should be also included (in brief) in the Abstract, as indicated in my previous remark. 

In the paragraphs 4.3 and 4.4 you need references to support the methodology you used for the experiment; the same also exists for the paragraph 4.6.

In the conclusions you need also some information on the future perspectives of the proposed fertilization strategy (i.e. how do you think Trichoderma applications will help the pepper producers to boost yields, improve fruit quality and nutrient uptake? is it economically viable for the farmers to apply these commercial products?)

Generally English quality is very low and the text needs extended and rigorous linguistic revision before it could be reconsidered. 

Recommendation to the Editor: Interesting topic, but the text needs remarkable major revision before it can be reconsidered.  

English quality is very low and the text needs to be extensively revised by an English native speaker, to improve its quality. 

Author Response

Dear Reviewer,

Thank you very much for your letter and advice on our manuscript. According to your suggestions, now we’ve thoroughly revised the manuscript. We hope that you can recognize the efforts that we’ve made on the improvement of this manuscript and are satisfied with this revised version. The specific revisions have been attached.

Best wishes!

Xiaoyu Duan

Reviewer 3 Report

This manuscript is not ready for review. It should have been sent back to the authors by the editors. Results are not presented in any comprehensive way. They are not tabulated. We cannot see them, only the bits the authors think important given in the results

This is very amateurish. You should returns to the authors without sending it to review.

Author Response

Dear Reviewer,

We appreciate that you spent your precious time on reading our manuscript and gave us constructive comments for this manuscript. In terms of the results that was not presented in the manuscript, it is necessary to explain the fact that we have arranged the important results into graphs and tables and uploaded them to the website when the manuscript was firstly submitted.

 Best wishes!

Xiaoyu Duan 

Round 2

Reviewer 2 Report

Despite the improvement in the quality of text, the style of data presentation still remains monotonous; only two Tables exist, without ant graph/figure inclusion. In addition, little discussion and comparison of the main results to those of other researchers exist; I would like to see more discussion on your main data...   

The other problem is the linguistic quality of the ms; although there are some significant improvements, it still needs efforts to ameliorate the meaning and to avoid grammatical errors. 

Finally, in the Conclusions, I would like to see some more suggestions/future perspectives about the economic viability for the farmers of Trichoderma applications.

Linguistic quality still remains below average; thus, further efforts are needed to improve the meaning and avoid grammatical errors

Author Response

Dear Reviewer,

Thank again for your constructive comments for this manuscript. According to your suggestion, we have revised the manuscript, and it was sent to the edigate language service company for further improving and correcting the grammatical errors. I hope that you could satisfy with this manuscript.

 Best wishes!

Xiaoyu Duan 

Reviewer 3 Report

Two aspects of this manuscript have been very unsatisfactory. One is handling. When it was first sent to me, it came without the figures. Without them, it was impossible to evaluate it. According to the author’s reply, the figures were submitted with the manuscript, but were not sent to me and presumably not to other reviewers. The manuscript was recently again sent to me but still without the figures. When I sent messages to your editorial office trying to obtain them, I got no response for many days. I sent an email directly to the corresponding author asking for the figures, but my messages were rejected as spam. Not much point being the corresponding author if you won’t accept correspondence. I have now at last received copies of the figures and can evaluate the data. And that is the other unsatisfactory aspect.

The essence of the manuscript is that there is an interaction between phosphorus supply and Trichoderma. If this were the case, there would be a positive response to P and this response would be steeper in the presence of Trichoderma. This is difficult to evaluate because the data are presented in columns (and we can’t see the effects of level very well), and because the essential comparison (plus or minus Trichoderma) and separated. The results can be better seen if point and line graphs are used as below.

This is not the response you would expect if the Trichoderma treatment was acting to spare phosphorus. It is also shown in the next graph.

These graphs tell us two things. One is that there is a benefit from Trichoderma, but it is not related to phosphate supply. There is a similar benefit no matter what the P level is. The other is that the “standard” treatment (the 100 level) is far too high. I do not know what that level is because their table 2 gives the superphosphate application in terms of grams but there is no indication of the area to which this was applied. It would be interesting to calculate the P balance – the P added minus the P removed in the crop, but this is not possible on the information given. An

The manuscript canvasses several ways in which Trichoderma could be beneficial, but these are not investigated, presumably because the authors think of the effects in terms of P supply. Without that investigation, all we have is an unexplained observation – expressed as “addition of Trichoderma could contribute to healthy growth in peppers” (line 244).

The most troubling aspect of this work is the consistent depression of growth at the highest level of P application. These trace back to the belief that previous applications of phosphate have become “fixed” and there is therefore a need to continually supply heavy doses. For example, “high amounts of phosphorus fertiliser induce the phosphorus to rapidly combined with aluminium, iron and calcium forming an unavailable complex before the plant roots take up the phosphorus” (line 55) Given this frame of reference it is not surprising that the authors think of P fertilisers as evil. I suggest the solution to this problem is to better understand the way the P actually interacts with soil and the effect of this legacy P on the availability of P. Some references are given at the bottom of this message.

We are given no information on the previous usage and P application of the site used, which again makes it difficult to evaluate the responses. I note that the P extracted by bicarbonate (their table 1) is 104 mg/kg. That is a high value. It is rather difficult to expect P response in the presence of such a high supply from soil. By the way, this quantity is wrongly described as “available P” in table 1. There is no such quantity. The number should always be described in terms of the procedure used – in this case presumably “Olsen P”.

Because I think this study is fundamentally flawed, I will not go into problems of expression.

My recommendation is that this manuscript cannot be accepted. I suggest to the authors that they might consider submitting a different manuscript. In it they might point out that Trichoderma does not spare phosphate but gives a similar response at all phosphate levels. It might also point out that poor understanding of soil phosphate chemistry leads to wasteful, and even dangerous, applications of P fertiliser.

1.       Soil phosphate chemistry and the P-sparing effect of previous phosphate applications. Plant and Soil. 397, 401 – 409.  DOI 10.1007/s11104-015-2514-5.

2.       How understanding soil chemistry can lead to better phosphate fertilizer practice: a 68 year journey (so far). Plant Soil https://doi.org/10.1007/s11104-022-05468-4

3.       Evaluating the benefits of legacy phosphate. Plant Soil (2022) 480:561–570   https://doi.org/10.1007/s11104-022-05601-3

It has improved

Author Response

Thanks again for your letter and thoughtful comment for improving this manuscript. According to your suggestion, we have revised the manuscript, and we hope that you can recognize our efforts in this manuscript.

 Best wishes!

Xiaoyu Duan 

Round 3

Reviewer 2 Report

The authors have seriously tried to ameliorate the quality of the presentation of their data, based on my previous remarks; I would like to congratulate them for their efforts. In addition, the information provided in the conclusions has been updated with some necessary future directions. 

I really believe that now their data merit publication, after some minor linguistic improvements.   

some minor linguistic improvements are still needed, before publication 

Author Response

Dear reviewer,

Sincerely thank for your comments and recognition of our effort in this manuscript. According to your suggestions, this manuscript was once again sent to language service company (editage) for correct language errors by professionals, and the revised sections were marked in red. We believe that this revised manuscript would make your satisfaction.

Best wishes!

Xiaoyu Duan

Reviewer 3 Report

Whether this article should be accepted by the journal “Plants”, depends on the journal’s policy. According to its website it is an “international scientific” journal. It is not a local extension journal, that is, a journal that publishes recipes that advise farmers how to best grow their crops. The submitted article advises farmers how best to combine phosphate treatments and Trichoderma treatments. It is made more so in the current draft where best combinations are emphasised. It is appropriate for farmers who choose to grow peppers in soils with similar phosphate history to the one used and in a similar environment.

If the manuscript were to be appropriate for a scientific journal, it would emphasise principles. It would present the data in a way to reflect those principles. I have indicated these in my first report. The authors have not accepted the suggestions and have made the manuscript even more of an extension report. That reinforces my opinion that the manuscript is not suitable for a scientific journal.

no comment

Author Response

Dear reviewer,

Thanks again for your comments and suggestion for this manuscript. The soil condition is complex. Due to the great difference of soil characteristics in different regions, no experiment is universal to resolve a specific problem. This experiment was conducted in Northeast China to resolve the problem that a large accumulations of residual phosphorus fertilizer in the soil in this area. It is worth mentioning that this novel application strategy in this study was widely used in Northeast China and make remarkable achievement. Additionally, this study provided a valuable reference for other regions with the same problems.

Best wishes!

Xiaoyu Duan